# Geometrothermodynamic Cosmology

**DOI:** 10.3390/e25071037

**Published:** 2023-07-10

**Authors:** Orlando Luongo, Hernando Quevedo

**Affiliations:** 1Scuola di Scienze e Tecnologie, Università di Camerino, Via Madonna delle Carceri 9, 62032 Camerino, Italy; 2SUNY Polytechnic Institute, Utica, NY 13502, USA; 3INFN, Sezione di Perugia, 06123 Perugia, Italy; 4INAF—Osservatorio Astronomico di Brera, 20121 Milano, Italy; 5NNLOT, Al-Farabi Kazakh National University, Al-Farabi av. 71, Almaty 050040, Kazakhstan; 6Instituto de Ciencias Nucleares, Universidad Nacional Autónoma de México, AP 70543, Ciudad de México 04510, Mexico; quevedo@nucleares.unam.mx; 7Dipartimento di Fisica and ICRANet, Università di Roma “La Sapienza”, 00185 Roma, Italy

**Keywords:** relativistic cosmology, fundamental equations, geometrothermodynamics, 05.70.Ce, 04.70.-s, 04.20.-q, 05.70.Fh

## Abstract

We review the main aspects of geometrothermodynamics, a formalism that uses contact geometry and Riemannian geometry to describe the properties of thermodynamic systems. We show how to handle in a geometric way the invariance of classical thermodynamics with respect to Legendre transformations, which means that the properties of the systems do not depend on the choice of the thermodynamic potential. Moreover, we show that, in geometrothermodynamics, it is possible to apply a variational principle to generate thermodynamic fundamental equations, which can be used in the context of relativistic cosmology to generate cosmological models. As a particular example, we consider a fundamental equation that relates the entropy with the internal energy and the volume of the Universe, and construct cosmological models with arbitrary parameters, which can be fixed to reproduce the main aspects of the inflationary era and the standard cosmological paradigm.

## 1. Introduction

Differential geometry has been applied in many branches of theoretical physics for more than one century. For instance, the idea that the gravitational interaction can be understood in terms of the curvature of an abstract Riemannian manifold was first proposed by Einstein in 1915. In fact, this idea can be considered as the basis of the principle “field strength = curvature”. Today, we know that this principle can be used to explore the properties of not only the gravitational field but also of all the four known interactions of Nature (for an introduction to this subject, see, for instance, [1]).

Indeed, in 1953, Yang and Mills [2] put forth an alternative formulation of the electromagnetic field. According to their proposal, the Faraday tensor can be understood as the curvature of a principal fiber bundle, where the base manifold is the Minkowski spacetime and the standard fiber is given by the symmetry group U(1).

This idea was generalized to include the case of the weak and strong interactions, which together with the electromagnetism are known as gauge interactions. It has been already well established (see, for instance, [1]) that the weak and strong interactions can be described in terms of the curvature of a principal fiber bundle with a four-dimensional Minkowski base manifold and the standard fiber SU(2) and SU(3), respectively. Notice that the construction of all these theories is based upon the existence of specific symmetries. So, the invariance under diffeomorphisms is essential for the formulation of gravity theories. In the case of gauge theories, the invariance with respect to the gauge groups U(1), SU(2), or SU(3) is a fundamental aspect of the corresponding theories.

Consider now thermodynamics. Broadly speaking, we can say that all the known interactions act between and inside the particles that constitute a thermodynamic system. Because the number of particles in a thermodynamic system is very large, it is not possible to study the properties and interactions of all the particles. Instead, it is necessary to apply methods of statistical physics to find the average values of the physical quantities of interest. In this case, it is possible to introduce the concept of thermodynamic interaction by using the standard statistical approach to thermodynamics, according to which all the physical properties of the system can be derived from the corresponding Hamiltonian that defines the partition function [3]. Then, the interaction between the particles of the system is described by the potential part of the Hamiltonian. Consequently, if the potential vanishes, we say that the system has no thermodynamic interaction. The question arises whether it is possible to formulate a geometric description of thermodynamics that takes into account its symmetry properties and follows Einstein’s principle that the thermodynamic interaction corresponds to the curvature of a manifold. We will see that the answer is affirmative in the context of geometrothermodynamics (GTD), a formalism in which the underlying symmetry corresponds to the Legendre transformations of classical thermodynamics.

However, the application of geometry in thermodynamics is not new.

In the realm of equilibrium thermodynamics, three branches of geometry have been extensively utilized: analytic geometry, Riemannian geometry, and contact geometry. These branches have played significant roles in advancing our understanding and analysis of thermodynamic systems at equilibrium.

In the case of analytic geometry, in the pioneer works by Gibbs, it was established that phase transitions can be represented as extremal points on the surface defined by the equations of state of the system. In fact, this idea was also included in the classification of phase transitions proposed by Ehrenfest (for a more detailed description of these contributions see, for instance, [4,5]). On the other hand, in 1945, Rao used in [6] the Fisher information matrix [7] to introduce a Riemannian metric in statistical physics and thermodynamics. The Fisher–Rao metric has been also used in the framework of information theory and statistics (see, e.g., [8] for a review). Furthermore, to describe the geometric properties of the equilibrium space, Weinhold [9,10,11] and Ruppeiner [12,13,14] proposed to use Hessian metrics, in which the Hessian potential is taken as the internal energy and (minus) the entropy, respectively. In general, these metrics have been used intensively to describe ordinary systems and black holes [15,16,17,18,19,20,21,22,23]. Finally, contact geometry was introduced by Hermann [24] to study the thermodynamic phase space and to formulate in a consistent manner the geometric version of the laws of thermodynamics [24].

The formalism of GTD distinguishes itself from the aforementioned approaches by its fundamental principle, which is the preservation of classical thermodynamics under the interchange of thermodynamic potentials, corresponding to Legendre transformations [25]. In pursuit of this principle, GTD incorporates nearly all previous developments, particularly the geometric concepts associated with phase and equilibrium spaces.

A primary objective of GTD is to provide an invariant interpretation of the curvature exhibited by the equilibrium space as a manifestation of thermodynamic interaction. Consequently, the equilibrium space of an ideal gas is expected to be described by a Riemannian manifold with zero curvature. However, in the case of systems with thermodynamic interaction, the Riemannian curvature should be non-zero, and phase transitions should correspond to critical points that hold significance within the equilibrium space. As we delve deeper into GTD, we will observe that all these intuitive conditions are duly considered.

The formalism of GTD has been applied to describe ordinary thermodynamic systems, such as the classical ideal gas [26], van der Waals systems [27], realistic gases [28], ideal quantum gases, and Bose–Einstein condensates as well [29]. In chemistry, we have shown that chemical reactions can be represented as geodesics of a Riemannian manifold called equilibrium space [30]. Additionally, in econophysics, it has been shown that certain economies can be interpreted as thermodynamic systems with phase transitions representing financial crises [31]. Several works have been dedicated to different aspects of GTD and to study the properties of black holes in different gravity theories [32,33,34,35,36,37,38,39,40,41,42].

In this review, our focus is on providing a thorough overview of GTD and delving into its core formalism. Specifically, we concentrate on its application within the context of relativistic cosmology [43,44,45,46], examining both the late and early stages of the Universe’s evolution. We particularly explore the key aspects of the fundamental equations in GTD that are relevant to cosmology, adopting the cosmological principles of homogeneity and isotropy on cosmic scales and treating the Universe as a thermodynamic system. We discuss how GTD can reproduce the well-established ΛCDM model, showcasing its ability to capture the evolution of dark energy from a fundamental equation. Additionally, we investigate the construction of a GTD inflationary fluid within these cosmological scenarios, which can replicate the outcomes traditionally obtained through the slow-roll approximation for fields. Throughout the review, we critically discuss the perspectives and limitations of our approaches within the framework of GTD, highlighting the expectations for future developments.

This work is organized as follows. In Section 2, we present the main ideas and goals of GTD. In particular, we introduce formally in Section 2.1 the concept of thermodynamic phase space, which is essential for the implementation of the invariance with respect to Legendre transformations of the thermodynamic potential. We also define in Section 2.2 the equilibrium space as a subspace of the phase manifold in which the laws of thermodynamics are valid and the geometric structure is determined for each thermodynamic system from the corresponding fundamental equation. Section 3 is devoted to the study of GTD in the case of systems with two thermodynamic degrees of freedom. We present the explicit form of the metrics of the phase and equilibrium spaces and calculate the points where curvature singularities occur. We also show that the singularities determine the locations where the system becomes unstable and phase transitions take place. Furthermore, in Section 4, we present the variational principle associated with harmonic maps and show that it leads to a set of differential equations, whose solutions can be interpreted as fundamental equations for thermodynamic systems. Then, in Section 5, we present a brief introduction to the standard cosmological model and explain the way how GTD can be incorporated into the framework of relativistic cosmology, which is the essence of geometrothermodynamic cosmology. Furthermore, in Section 6, we present the details of a particular geometrothermodynamic cosmological model. Finally, in Section 7, we discuss our results. Throughout this paper, we use units in which $G=c=k_{{}_{B}}=\hbar =1$.

## 2. The Formalism of GTD

First, let us introduce some notations and conventions that we will use throughout this work. In equilibrium thermodynamics, to describe a system with *n* thermodynamic degrees of freedom, we use a thermodynamic potential Φ, *n* extensive variables $E^{a}$ (a=1,2,⋯,n) and *n* intensive variables $I^{n}$. All the properties of the system can be derived from the fundamental equation $\Phi =\Phi (E^{a})$, which is assumed to satisfy the first law of thermodynamics,
(1)$$d\Phi =\sum_{a=1}^{n}I_{a}dE^{a}\;,\;I_{a}=\frac{\partial \Phi }{\partial E^{a}}\;.$$
Usually, Φ is taken as the entropy *S* or the internal energy *U* of the system, choices that lead to the entropic and energetic representations, respectively. In addition, the potential Φ, as a function of the extensive variables $E^{a}$, is also assumed to satisfy the second law of thermodynamics so that the fundamental equation $\Phi =\Phi (E^{a})$ contains all the physical information of the corresponding thermodynamic system.

The main point of the first law is that it allows us to write down explicitly the equations of state $I_{a}=I_{a}\left(E^{a}\right)$, which essentially determine all the thermodynamic properties of the system. An important property of classical thermodynamics is that it does not depend on the choice of thermodynamic potential. Indeed, from the potential Φ, we can obtain new potentials $\tilde{\Phi }$ by using the Legendre transformation
(2)$$\tilde{\Phi }=\Phi +\sum_{i=1}^{j}I_{i}E^{i}=\Phi +\sum_{i=1}^{j}\frac{\partial \Phi }{\partial E^{i}}E^{i}\;,$$
where *j* is any integer of the set {1,2,⋯,n}. If j=n, the Legendre transformation is called total; otherwise, it is called partial. A well-known fact of classical equilibrium thermodynamics is that the properties of a system do not depend on the choice of the potential $\tilde{\Phi }$ [5]. Recall that this property is valid only at the level of classical equilibrium thermodynamics. The situation at the level of statistical thermodynamics is different because, in this case, the properties of a statistical system can depend on the choice of ensemble. To consider this dependence it would be necessary to construct the corresponding formalism of statistical GTD. At the moment, however, the situation is such that no statistical ensembles can be defined at the level of the equilibrium space of GTD, which is a Riemannian manifold, whose points represent equilibrium states. Probably, statistical GTD will require the use of statistical Riemannian manifolds, in which each point represents a probability distribution [47]. This is a problem that we expect to consider in the future.

To provide a better understanding, it is important to note that within a given ensemble, different thermodynamic quantities can be used to describe the system. Despite the choice of potential, the measurements obtained from these quantities should yield consistent results. Therefore, thermodynamics can be described irrespective of the specific potential employed. However, it is worth mentioning that changing the ensemble can lead to alternative results that may appear different. An illustrative example is the behavior of black holes in AdS. In the grand-canonical ensemble, a Hawking–Page phase transition occurs, while in the canonical ensemble, a van der Waals-like phase transition can take place. The distinction arises from the fact that Legendre invariance is not initially imposed on the geometric theory, but rather applied later through Legendre transformations on the potentials. In the geometric approach to thermodynamics (GTD), on the other hand, Legendre transformations are imposed from the outset due to the underlying geometric nature of the theory. This ensures that the fundamental principles of equilibrium thermodynamics are incorporated into a geometric framework that remains invariant under Legendre transformations.

This is crucial to ensure that the choice of thermodynamic potential does not alter the system’s properties. However, traditional Legendre transformations cannot be treated as simple coordinate transformations since they involve derivatives of the potential. To address this issue, we propose a solution by considering all variables, including Φ, $E^{a}$, and $I_{a}$, as independent coordinates. In this approach, Legendre transformations are represented as algebraic relationships between these coordinates.

To describe this procedure, let us consider a set of 2n+1 coordinates denoted as $Z^{A}=\left(\Phi ,E^{a},I_{a}\right)$, where *A* ranges from 0 to 2n. Then, a Legendre transformation can be defined as a coordinate transformation of the form: $Z^{A}\to {\tilde{Z}}^{A}=\left(\tilde{\Phi },{\tilde{E}}^{a},{\tilde{I}}^{a}\right)$ such that [48]
(3)$$\Phi =\tilde{\Phi }-\delta_{kl}{\tilde{E}}^{k}{\tilde{I}}^{l}\;,\;E^{i}=-{\tilde{I}}^{i},\;\;E^{j}={\tilde{E}}^{j},\;I^{i}={\tilde{E}}^{i},\;\;I^{j}={\tilde{I}}^{j}\;,$$
where i∪j is any disjoint decomposition of the set of indices {1,⋯,n}, and k,l=1,⋯,i.

Specifically, when i=∅, the transformation described in Equation (Equation 3) corresponds to the identity transformation. On the other hand, for i=1,⋯,n, Equation (Equation 3) represents a total Legendre transformation. Here, we denote $I^{a}=\delta^{ab}I_{b}$, where $\delta^{ab}=\mathrm{diag}\left(1,\cdots ,1\right)$, and we adopt the convention of summation over repeated indices for simplicity. It is evident that if we introduce the explicit dependence $I_{a}=\frac{\partial \Phi }{\partial E^{a}}$, the transformation (Equation 3) reduces to (Equation 2). Furthermore, it can be easily shown that the Jacobian of the Legendre transformation is non-zero, indicating the existence of an inverse transformation. This implies that we have represented Legendre transformations in the phase space as a diffeomorphism.

Now, we can use the Legendre transformation (Equation 3) to introduce another important geometric structure of GTD.

### 2.1. The Phase Space

Let T be a (2*n* + 1)-dimensional space with coordinates $Z^{A}$. Then, Darboux theorem states that, in T, there exists a canonical 1-form
(4)$$\Theta =d\Phi -I_{a}dE^{a}\;,$$
which satisfies the condition
(5)$$\Theta \wedge {\left(d\Theta \right)}^{n}\neq 0\;,$$
and is called contact form. The pair (T,Θ) is known as a contact manifold. The main point of this canonical construction is that the contact 1-form Θ is invariant with respect to transformations in the sense that, under the coordinate transformation (Equation 3), it behaves as
(6)$$\Theta =d\Phi -I_{a}dE^{a}\to \tilde{\Theta }=d\tilde{\Phi }-{\tilde{I}}_{a}d{\tilde{E}}^{a}\;.$$

We can state that a geometric quantity is Legendre invariant whether it transforms as Equation (Equation 6) by virtue of a Legendre transformation.

We define the phase space of GTD as the triad (T,Θ,G), where $G=G_{AB}dZ^{A}dZ^{B}$ is a line element with a Riemannian metric $G_{AB}=G_{AB}\left(Z^{C}\right)$, which should satisfy the Legendre invariance condition
(7)$$G=G_{AB}dZ^{A}dZ^{B}\to \tilde{G}={\tilde{G}}_{AB}d{\tilde{Z}}^{A}d{\tilde{Z}}^{B}\;.$$

Notice that Equations (Equation 6) and (Equation 7) are the symbolic representation of what we mean under Legendre invariance. In fact, this is the analog of the form invariance required for the Minkowski under Lorentz transformations or of the Hamilton equations under canonical transformations.

As we can see, the goal of introducing the metric *G* is to take into consideration Legendre invariance in terms of coordinate transformation. This represents its importance for the geometric construction we are dealing with. From a physical point of view, however, the metric structure *G* is necessary to ensure that the physical properties of a given system do not depend on the choice of the thermodynamic potential. Indeed, as we will see later, the metric *G* induces a new metric *g* on the equilibrium space, where the laws of thermodynamics are satisfied and the thermodynamic system exist. In the context of geometric approaches to thermodynamics, the metric *g* represents a thermodynamic metric rather than a purely geometric one. It is a mathematical object where the coordinates are replaced by thermodynamic quantities, allowing for a different interpretation of the underlying space compared to traditional geometric spacetimes in general relativity.

The tilde transformation shown above is the representation of the Legendre transformation in terms of coordinates. This is significant because in GTD the above coordinate transformations correspond to Legendre transformations that act on the variables of the metric. These transformations provide a physical interpretation of the metric itself, which is different from other geometric approaches to thermodynamics, where the potentials are not variables of the metric.

By incorporating Legendre transformations into the geometric formalism, the approach of GTD aims to unify equilibrium thermodynamics with a geometric framework that remains invariant under such transformations. This allows for a deeper understanding of the thermodynamic properties of a system within a geometric context.

As explained above, the components of the metric $G_{AB}$ should maintain their functional dependence on $Z^{A}$ under a Legendre transformation (Equation 3). When expressed explicitly, this condition gives rise to a system of algebraic equations [25], which restrict the dependence of the components $G_{AB}$ on the coordinates $Z^{A}$. A thorough analysis of this system reveals that the solutions can be classified into three distinct classes of metrics, which can be expressed as follows:(8)$$G^{{}^{I}}={\left(d\Phi -I_{a}dE^{a}\right)}^{2}+\left(\xi_{ab}E^{a}I^{b}\right)\left(\delta_{cd}dE^{c}dI^{d}\right)\;,$$
(9)$$G^{{}^{II}}={\left(d\Phi -I_{a}dE^{a}\right)}^{2}+\left(\xi_{ab}E^{a}I^{b}\right)\left(\eta_{cd}dE^{c}dI^{d}\right)\;,$$
(10)$$G^{III}={\left(d\Phi -I_{a}dE^{a}\right)}^{2}+\sum_{a=1}^{n}\xi_{a}{\left(E_{a}I_{a}\right)}^{2k+1}dE^{a}dI^{a}\;,$$
where $\delta_{ab}=\mathrm{diag}\left(1,1,\cdots ,1\right)$, $\eta_{ab}=\mathrm{diag}\left(-1,1,\cdots 1\right)$, $\xi_{a}$ are real constants, $\xi_{ab}$ is a diagonal n×n real matrix, and *k* is a **real** integer.

It turns out that the condition of Legendre invariance does not fix completely the form of the metric components $G_{AB}$ but leaves the coefficients *k*, $\xi_{a}$, and $\xi_{ab}$ arbitrary. We notice that the above metrics were derived under different conditions. Indeed, the metrics $G^{I}$ and $G^{II}$ are invariant with respect to total Legendre transformations, whereas the metric $G^{III}$ is also invariant under partial transformations.

We see that the main goal of the phase space is to incorporate the Legendre transformations of classical thermodynamics into the formalism as a particular diffeomorphism that relates the coordinates $Z^{A}$. The triad (T,Θ,G) constitutes a contact Riemannian manifold in which the contact 1-form Θ and the metric structure *G* are Legendre invariant. Thus, we have reached one of the main goals of GTD, which consists in constructing a formalism that contains Legendre invariance as one of the main elements.

### 2.2. The Equilibrium Space

In GTD, we define the *n*-dimensional equilibrium space E as the set of points in which a thermodynamic system with *n* degrees of freedom can be in equilibrium. Each point in E represents an equilibrium state of the system. In order for E to possess the same properties as the phase space T, we define it as a subspace of T generated by the smooth embedding map:(11)φ:E⟶T
or in coordinates
(12)$$\varphi :\left\{E^{a}\right\}⟼\left\{\Phi \left(E^{a}\right),E^{a},I_{a}\left(E^{a}\right)\right\}$$
such that the condition
(13)$$\varphi^{*}\left(\Theta \right)=\varphi^{*}\left(d\Phi -I_{a}dE^{a}\right)=0\;\;i.e.,\;\;d\Phi =I_{a}dE^{a}\;\;\mathrm{on}\;\;E$$
is satisfied, where $\varphi^{*}$ is the pullback of φ. Notice that the embedding map φ demands that the fundamental equation $\Phi =\Phi (E^{a})$ must be given explicitly in order for the equilibrium space to be well defined. This means that the geometric properties of E depend explicitly on the properties of the corresponding thermodynamic system. Notice also that the condition $\varphi^{*}\left(\Theta \right)=0$ is equivalent to saying that the first law of thermodynamics is valid on E. Then, the mere definition of the equilibrium space endows it with a fundamental equation $\Phi (E^{a})$ that satisfies the first law $d\Phi =I_{a}dE^{a}$, and since we also demand that $\Phi (E^{a})$ satisfies the second law, it follows that E is the space that should reflect all the properties of the thermodynamic systems.

Furthermore, the pullback $\varphi^{*}$ can be used to induce a metric *g* on E from the metric *G* of T by means of the relationship
(14)$$g=\varphi^{*}\left(G\right)\;$$
or in components
(15)$$g_{ab}=\frac{\partial Z^{A}}{\partial E^{a}}\frac{\partial Z^{B}}{\partial E^{b}}G_{AB}\;.$$

It follows then that each of the Legendre invariant metrics *G* of T induces its own metric on E.

Before deriving the explicit form of the metrics for the equilibrium space, it is important to analyze the properties of the fundamental equation $\Phi (E^{a})$. In ordinary thermodynamics, the fact that Φ is a function of the extensive variables implies that it should be a homogeneous function of degree one. This means that when the extensive variables are rescaled as $E^{a}\to \lambda E^{a}$, the function Φ exhibits the following behavior [49,50]
(16)$$\Phi \left(\lambda E^{a}\right)=\lambda \Phi \left(E^{a}\right),$$
where λ is a positive constant, i.e., it is a homogeneous function of degree one. However, there are systems whose fundamental equations are not homogeneous. For instance, the Hawking–Bekenstein relationship
(17)$$S=\frac{1}{4}A_{h}\;,$$
where *S* is the entropy of the black hole and $A_{h}$ is the area of the event horizon, does not satisfy the homogeneity condition (Equation 16). However, the experience shows that it satisfies the quasi-homogeneity condition [51,52,53]
(18)$$\Phi \left(\lambda^{\beta_{a}}E^{a}\right)=\lambda^{\beta_{\Phi }}\Phi \left(E^{a}\right)\;,$$
where $\beta_{a}$ and $\beta_{\Phi }$ are constants. Moreover, one can show that, in the case of quasi-homogeneous systems, the Euler identity can be written as
(19)$$\beta_{\Phi }\Phi =\beta_{ab}I^{a}E^{b},\;\mathrm{with}\;\beta_{ab}=\mathrm{diag}\left(\beta_{1},\beta_{2},\cdots ,\beta_{n}\right)\;.$$

We want to incorporate this property of quasi-homogeneous systems into the formalism of GTD by demanding that the explicit form of the metric *g* can be applied indistinguishably to homogeneous and quasi-homogeneous systems [54,55]. It turns out that this requirement fixes the form of the constants $\xi_{a}$ and $\xi_{ab}$, which enter the metrics of the phase space as given in Equations (Equation 8)–(Equation 10), as follows:(20)$$\xi_{a}=\beta_{a},\;\xi_{ab}=\beta_{ab}=\mathrm{diag}\left(\beta_{1},\beta_{2},\cdots ,\beta_{n}\right)\;.$$

Taking into account these conditions and the modified form of the Euler identity (Equation 19), the result of applying the pullback $\varphi^{*}$ on the metrics (Equation 8), (Equation 9), and (Equation 10) leads to the following metrics for the equilibrium space
(21)$$g_{ab}^{I}=\beta_{\Phi }\Phi \delta_{a}^{\;c}\frac{\partial^{2}\Phi }{\partial E^{b}\partial E^{c}},$$
(22)$$g_{ab}^{II}=\beta_{\Phi }\Phi \eta_{a}^{\;c}\frac{\partial^{2}\Phi }{\partial E^{b}\partial E^{c}},$$
(23)$$g^{III}=\sum_{a=1}^{n}\beta_{a}{\left(\delta_{ad}E^{d}\frac{\partial \Phi }{\partial E^{a}}\right)}^{2k+1}\delta^{ab}\frac{\partial^{2}\Phi }{\partial E^{b}\partial E^{c}}dE^{a}dE^{c}\;,$$
respectively, where $\delta_{a}^{\;c}=\mathrm{diag}\left(1,\cdots ,1\right)$, $\eta_{a}^{\;c}=\mathrm{diag}\left(-1,1,\cdots ,1\right)$.

We can observe that a given fundamental equation $\Phi (E^{a})$ yields three distinct metrics for the equilibrium space, all of which should accurately describe the properties of the same thermodynamic system. The explanation above demonstrates that GTD involves contact geometry at the phase space level T and Riemannian geometry at the equilibrium space level E. The entire geometric structure of GTD is well-defined from a mathematical perspective. In fact, it can be represented as shown in the diagram depicted in Figure 1. The diagram illustrates the relationship between T and E through the map φ, which, in turn, induces a pushforward $\varphi_{*}$ and a pullback $\varphi^{*}$ that operate between the corresponding tangent spaces $T_{E}$ and $T_{T}$, as well as their duals $T_{E}^{*}$ and $T^{*}T$. Furthermore, the diagram demonstrates how the metric of the phase space *G* is connected to the metric of the equilibrium space *g* via the pullback operation.

## 3. Two-Dimensional GTD

Immediately, it is possible to work out systems of interest that appear particularly interesting. For example, in the case of a thermodynamic system with n=2, i.e., a system with fundamental equation
(24)$$\Phi =\Phi (E^{1},E^{2}),$$
the metrics of the equilibrium space (Equation 21)–(Equation 23) can be written explicitly as
(25)$$\begin{array}{ccc} g^{I} & = & \beta_{\Phi }\Phi \left[\Phi_{,11}{\left(dE^{1}\right)}^{2}+2\Phi_{,12}dE^{1}dE^{2}+\Phi_{,22}{\left(dE^{2}\right)}^{2}\right]\; \end{array}$$
(26)$$\begin{array}{ccc} g^{II} & = & \beta_{\Phi }\Phi \left[-\Phi_{,11}{\left(dE^{1}\right)}^{2}+\Phi_{,22}{\left(dE^{2}\right)}^{2}\right]\;, \end{array}$$
(27)$$\begin{array}{ccc} g^{III} & = & \beta_{1}{\left(E^{1}\Phi_{,1}\right)}^{2k+1}\Phi_{,11}{\left(dE^{1}\right)}^{2}++\beta_{2}{\left(E^{2}\Phi_{,2}\right)}^{2k+1}\Phi_{,22}{\left(dE^{2}\right)}^{2}\\ & & +\left[\beta_{1}{\left(E^{1}\Phi_{,1}\right)}^{2k+1}+\beta_{2}{\left(E^{2}\Phi_{,2}\right)}^{2k+1}\right]\Phi_{,12}dE^{1}dE^{2}\;, \end{array}$$
where $\Phi_{,a}=\frac{\partial \Phi }{\partial E^{a}}$, etc.

In analogy to Riemannian geometry, some of the fundamental properties of a given metric can be developed by means of curvature invariants. In the case under consideration, it is sufficient to analyze the Ricci scalar. (It is also possible to evaluate higher-order invariants; however, in the case of two-dimensional manifolds there is only one independent component of the curvature, and so all higher-order curvature invariants are proportional to the Ricci scalar. On the other hand, we could also analyze non-curvature scalars. In the context of black holes, however, we have not found so far any interesting interpretation of coordinate singularities. Nevertheless, GTD has been also applied in the context of mathematical chemistry to represent chemical reactions as geodesics of the equilibrium space [30]. In this case, it turned out that coordinate singularities represent the conditions under which chemical equilibrium is reached.)

Consequently, we require that the singularities of $g^{III}$ are connected to those of $g^{I}$ and $g^{II}$ in a manner that allows all metrics to describe the same system. This condition leads to a specific value for the integer *k* in the metric $g^{III}$, namely, k=0. A straightforward computation of the first-order curvature scalars i.e., the Ricci scalars, yields
(28)$$R^{I}=\frac{N^{I}}{D^{I}}\;,\;\;D^{I}=2\beta_{\Phi }\Phi^{3}{\left[\Phi_{,11}\Phi_{,22}-{\left(\Phi_{,12}\right)}^{2}\right]}^{2}\;,$$
(29)$$R^{II}=\frac{N^{II}}{D^{II}}\;,\;\;D^{II}=2\beta_{\Phi }\Phi^{3}{\left(\Phi_{,11}\Phi_{,22}\right)}^{2}\;,$$
(30)$$R^{III}=\frac{N^{III}}{D^{III}}\;,\;\;D^{III}={\left[\beta_{\Phi }^{2}\Phi^{2}{\left(\Phi_{,12}\right)}^{2}-4\beta_{1}\beta_{2}E^{1}E^{2}\Phi_{,1}\Phi_{,2}\Phi_{,11}\Phi_{,22}\right]}^{3}\;,$$
respectively, where we have used the Euler identity
(31)$$\beta_{1}E^{1}\Phi_{,1}+\beta_{2}E^{2}\Phi_{,2}=\beta_{\Phi }\Phi \;,$$
to reduce the form of the function $D^{III}$. The functions $N^{I}$, $N^{II}$ and $N^{III}$ depend on Φ and its derivatives. Consequently, the singularities of the equilibrium space metrics are determined by the zeros of the functions $D^{I}$, $D^{II}$, and $D^{III}$. The condition $D^{I}=0$ implies that
(32)$$\Phi_{,11}\Phi_{,22}={\left(\Phi_{,12}\right)}^{2}$$
so that $D^{II}\neq 0$ and
(33)$$D^{III}={\left(\Phi_{,12}\right)}^{6}{\left[\beta_{\Phi }^{2}\Phi^{2}-4\beta_{1}\beta_{2}E^{1}E^{2}\Phi_{,1}\Phi_{,2}\right]}^{3}\;.$$
The expression inside the parenthesis is zero only if Φ depends on one variable, which is equivalent to setting $\Phi_{,12}=0$. Furthermore, the condition $D^{II}=0$, i.e, $\Phi_{,11}=0$ or $\Phi_{,22}=0$, implies that $D^{I}$ and $D^{III}$ are zero only for $\Phi_{,12}=0$.

We conclude that all the singularities are determined by the zeros of the second-order derivatives of Φ, namely,
(34)$$\begin{array}{ccc} I: & & \Phi_{,11}\Phi_{,22}-{\left(\Phi_{,12}\right)}^{2}=0\;, \end{array}$$
(35)$$\begin{array}{ccc} II: & & \Phi_{,11}\Phi_{,22}=0\;, \end{array}$$
(36)$$\begin{array}{ccc} III: & & \Phi_{,12}=0\;. \end{array}$$
The singularity *I* is related to the stability condition of a system with two degrees of freedom [5], which is usually associated with a first-order phase transition. Furthermore, the singularities II and III can be associated with second-order phase transitions. To show this explicitly, recall that the response functions of a thermodynamic system define second-order phase transitions and are essentially determined by the behavior of the independent variables $E^{a}$ in terms of their duals $I_{a}$, i.e.,
(37)$$C^{ab}=\frac{\partial E^{a}}{\partial I_{b}}=\frac{1}{\Phi_{,ab}},$$
which is obtained by using the definition $I_{b}=\Phi_{,b}$. Consequently, the zeros of the second-order derivatives of Φ can be associated with second-order phase transitions. Some examples of the application of the above procedure to determine the phase transition structure of homogeneous systems can be consulted in [27,28].

## 4. Generating Fundamental Equations

Typically, the fundamental equation of a thermodynamic system is obtained through the analysis of empirical equations of state. This approach is commonly employed in the study of ordinary systems in the fields of chemistry and experimental physics. Another approach is to postulate fundamental equations, as is in the case of black holes. The formalism of GTD presents an alternative method. It utilizes harmonic maps, which we can apply to the embedding map connecting the equilibrium and phase spaces.

The embedding map φ:E⟶T has been utilized in the previous section to define the space of equilibrium states in a manner that naturally incorporates the first law of thermodynamics and the conditions for thermodynamic equilibrium. The pullback of this map is also employed to establish a relationship between the Legendre invariant metrics in T and E. As both spaces are equipped with Riemannian metrics, we can apply a specific variational principle as follows. Consider the phase space (T,Θ,G) and suppose that an arbitrary non-degenerate metric *h* is given in E with coordinates $E^{a}$. The smooth map φ:E⟶T, or in coordinate form $\varphi :E^{a}⟼Z^{a}$, is referred to as a *harmonic map* if the coordinates $Z^{A}$ satisfy the differential equations derived from the variation of the action [56].
(38)$$I_{h}=\int_{E}d^{n}E\;\sqrt{\left|h\right|}\;\;h^{ab}Z_{,a}^{A}Z_{,b}^{B}G_{AB}\;,$$
where $\left|h|=|\det(h_{ab})\right|$. The computation of the variational derivative with respect to $Z^{A}$ leads to
(39)$$\frac{\delta I_{h}}{\delta Z^{A}}=0\Leftrightarrow {□}_{h}Z^{A}:=\frac{1}{\sqrt{\left|h\right|}}{\left(\sqrt{\left|h\right|}\;h^{ab}Z_{,a}^{A}\right)}_{,b}+\Gamma_{\;BC}^{A}Z_{,b}^{B}Z_{,c}^{C}h^{bc}=0\;$$
where $\Gamma_{\;BC}^{A}$ are the Christoffel symbols associated to the metric $G_{AB}$, i.e.,
(40)$$\Gamma_{\;BC}^{A}=\frac{1}{2}G^{AD}\left(G_{DB,C}+G_{DC,B}-G_{BC,D}\right)\;.$$
For given metrics *G* and *h*, this is a set of 2n+1 second-order partial differential equations for the 2n+1 thermodynamic variables $Z^{A}$. These equations are under the form of the so-called Nambu–Goto equations [57]. They have been obtained directly from the action; therefore, their form appears independent of the particular hypothesis made on the metric. In other words, they remain invariant in form, with different dynamics imposed by selecting a given metric.

Moreover, the variation of the action (Equation 38) with respect to the metric $h_{ab}$ determines the “energy-momentum” tensor
(41)$$\frac{\delta I_{h}}{\delta h^{ab}}=0\Leftrightarrow T_{ab}:=g_{ab}-\frac{1}{2}h_{ab}h^{cd}g_{cd}=0\;,$$
where $g_{ab}$ is the metric induced on E by the pullback $\varphi^{*}$ according to (Equation 15). This algebraic constraint relates the metric components $h_{ab}$ with the components of the induced metric $g_{ab}$. From the last equation it is easy to derive the expression [57]
(42)$$h^{ab}g_{ab}=2{\left(\frac{\left|g\right|}{\left|h\right|}\right)}^{1/2}\;,$$
where $\left|g|=|\det(g_{ab})\right|$.

There is an equivalent description in terms of a Nambu–Goto-like action. Introducing the relationship (Equation 42) into the action (Equation 38) and using the expression (Equation 15) for the induced metric, we obtain the action
(43)$$I_{g}=2\int_{E}d^{n}E\;\sqrt{\left|g\right|}\;,$$
from which we derive the Nambu–Goto equations
(44)$${□}_{g}Z^{A}=\frac{1}{\sqrt{\left|g\right|}}{\left(\sqrt{\left|g\right|}\;\;g^{ab}Z_{,a}^{A}\right)}_{,b}+\Gamma_{\;BC}^{A}Z_{,b}^{B}Z_{,c}^{C}g^{bc}=0\;.$$
Here, instead of the arbitrary metric *h* we have the induced metric *g* so that if we specify the metric *G*, the induced metric is also fixed, and the resulting equations involve only the thermodynamic variables $Z^{A}$. Since the action $I_{g}$ is proportional to the volume element of the manifold E, the Nambu–Goto equations (Equation 44) can be interpreted as stating that the volume element induced in E is an extremal in T.

The above variational principle is based upon the use of harmonic maps, in which the scalar quantity entering the action is constructed with the determinants of the metrics *G* and *g*, only [57]. However, it is possible to introduce many other variational principles, in which the action could depend on other quantities such as the curvature scalar of *G* and *g*, higher-order curvature scalars, combinations within them, etc. So far, in the search for fundamental equations, we have not found new information in other variational principles, which include only first-order curvature invariants.

Equation (Equation 44) are highly non-trivial. Indeed, if we consider the component $Z^{0}=\Phi$ and recall that on E the thermodynamic potential Φ is a function of the extensive variables $E^{a}$, Equation (Equation 44) leads to a second order differential equation for Φ. For the components A=1,⋯,n, we have that $Z_{,b}^{A}=E_{,b}^{a}=\delta_{b}^{a}$ and equations (Equation 44) reduce to a set of first-order differential equations for the components $g^{ab}$, which include first-order derivatives of $G_{AB}$ and of $Z^{A}$. Finally, if we consider the components A=n+1,⋯,2n, then $Z_{,b}^{A}=Z_{,b}^{n+a}=I_{,b}^{a}=\Phi_{,ab}$. Then, Equation (Equation 44) reduces to a set of third-order differential equations for Φ, which are closely related to the set of second-order differential equations obtained for A=0. In other words, the fact that the harmonic map φ:E⟶T transforms the coordinates $Z^{A}$ into scalar functions of $E^{a}$, satisfying the differential relations given by the equilibrium conditions
(45)$$I_{a}=\frac{\partial \Phi }{\partial E^{a}}$$
increases the complexity of the Nambu–Goto equations. Moreover, the fact that the background metric $G_{AB}$ in GTD is always a curved metric that represents an additional problem. In spite of these difficulties, we will see below that it is possible to find exact solutions for the Nambu–Goto equations. Indeed, consider the case of two-dimensional equilibrium space (E,g) with the particular metric $g^{III}$ as given in Equation (27). The corresponding five-dimensional metric *G* of the phase space can be easily calculated from the general expression (Equation 10). To be in agreement with the result we obtained for the metric (27), we fix the constants as k=0 and $\xi_{a}=\beta_{a}$. Then, we obtain
(46)$$G^{III}={\left(d\Phi -I_{1}dE^{1}-I_{2}dE^{2}\right)}^{2}+\beta_{1}\left(E_{1}I_{1}\right)dE^{1}dI^{1}+\beta_{2}\left(E_{2}I_{2}\right)dE^{2}dI^{2}\;.$$
If we insert now (27) and (Equation 46) into the Nambu–Goto Equation (Equation 44), we obtain a set of five second-order differential equations for $Z^{A}=\left(\Phi ,E^{1},E^{2},I_{1},I_{2}\right)$. The equations for $E^{1}$ and $E^{2}$ turn out to be satisfied identically, whereas the remaining three equations constitute a system of differential equations for Φ, for which we obtained the following particular solutions:(47)$$\Phi =c_{1}{\left(E^{1}\right)}^{\alpha }{\left(E^{2}\right)}^{\beta }\;,$$
(48)$$\Phi =c_{1}\ln\left[{\left(E^{1}\right)}^{\alpha }+c_{2}{\left(E^{2}\right)}^{\beta }\right]\;,$$
(49)$$\Phi =c_{1}\ln\left(E^{1}+\frac{\alpha }{E^{2}}\right)+c_{2}\ln\left(E^{2}-\beta \right)\;,$$
where $c_{1},\;c_{1},\;c_{2},\;\alpha$, and β are constants. Essentially, the above solutions are fundamental equations and the question arises whether they can be used to describe realistic systems. The answer to this question is affirmative, and in particular, it has been shown that the first two solutions can be applied in the context of cosmology to construct unified models of dark matter and dark energy [43]. In Section 6, we will show that the third solution can be used to reproduce the standard cosmological model with an inflationary component.

The above fundamental equations clearly can be adapted to any particular physical system. Since the corresponding forms are general, there is no intuitive reason to exclude them or to predict their physical applicability a priori to describe the evolution of the Universe. Only a posteriori one can argue that, by virtue of the aforementioned choices, it is also possible to reproduce the effects of exotic fluids, whose equations of state seem to indicate repulsive effects in gravity. In this respect, it appears impossible for standard thermodynamics to exclude the above solutions that, instead, are quite evident only after exploiting the aforementioned relations. It appears natural, therefore, to work out those solutions in relation to dark matter and dark energy. Additionally, as an intriguing possibility, one can envision the universe as a brane existing within a higher-dimensional thermodynamic phase space, drawing an analogy to string theories. This analogy has been pointed out in [58].

## 5. Relativistic Cosmology

The objective of cosmology is the study of the Universe. In general, to study the Universe it is necessary to consider all the interactions known in Nature. However, at different scales, each interaction plays a different role. We are interested in describing the Universe at large scales, i.e., at the scale of hundreds of megaparsecs, which corresponds to about $10^{8}$ light years. At these scales, the galactic clusters can be considered as points so that the internal structure of the clusters, galaxies, etc., can be neglected. Moreover, at large scales, the dominant interaction is gravity, and the distribution of clusters in the Universes can be considered as homogeneous and isotropic, according to observations.

To construct a cosmological model at large scales, we start from several assumptions that we suppose to be valid during the entire evolution of the Universe. We formulate these assumptions as follows:The Universe is homogeneous and isotropic at each instant of time.Gravity is the dominant interaction of the Universe, and its behavior is dictated by Einstein’s equations [59]
(50)$$R_{\mu \nu }-\frac{1}{2}g_{\mu \nu }R=8\pi T_{\mu \nu }\;,$$
where $R_{\mu \nu }$ is the Ricci tensor, $g_{\mu \nu }$ is the metric tensor of the spacetime of the Universe, *R* the curvature scalar, and $T_{\mu \nu }$ is the energy-momentum tensor of the Universe.At large scales, the Universe can be considered as a perfect fluid with energy-momentum tensor [60]
(51)$$T_{\mu \nu }=\left(\rho +p\right)u_{\mu }u_{\nu }+pg_{\mu \nu }\;,$$
where ρ is the density, *p* the pressure, and $u_{\mu }$ is the 4-velocity of the observer, which we assume to move with the particles of the fluid.The Universe can be considered as a thermodynamic system.

The first three assumptions are standard and are mentioned in different forms in most textbooks. However, the fourth assumption is usually not mentioned explicitly, but we need to assume it in this work in order to be able to apply the formalism of GTD. The problem with the assumption that the Universe is a thermodynamic systems, i.e., a system in which the laws of thermodynamics are valid, is that according to the standard approach to classical thermodynamics the Universe needs to be in contact with a thermal reservoir. In this case, it is not clear where the reservoir could be since the system occupies the entire Universe. However, one can assume that the Universe is an isolated system to avoid some conceptual issues of thermodynamics. In any case, the fourth assumption is a controversial issue. Nevertheless, we suppose its validity and proceed as it is often assumed in theoretical studies: If the resulting model is physically meaningful, the starting assumptions should also be physical, at least to the extent of validity of the model.

We proceed now to construct a cosmological model based on the above assumptions. The first assumption is used to fix the form of the underlying metric $g_{\mu \nu }$. Indeed, homogeneity and isotropy are symmetries that can be implemented into the structure of the spacetime metric by using standard methods of differential geometry. The result is known as the Friedmann–Lemaître-Robertson–Walker line element, which, in polar coordinates (t,r,θ,ϕ), reads
(52)$$ds^{2}=-dt^{2}+a^{2}\left(t\right)\left[\frac{dr^{2}}{1-kr^{2}}+r^{2}\left(d\theta^{2}+\sin^{2}\theta d\phi^{2}\right)\right]\;,$$
where a(t) is the scale factorm and k=0,±1 is a constant that represents the constant spatial (t=0) curvature of this spacetime.

We now apply Assumptions 2 and 3. Einstein’s equations (Equation 50) for the metric (Equation 52) with energy-momentum (Equation 51) can be written as the Friedmann equations
(53)$$\frac{{\dot{a}}^{2}}{a^{2}}+\frac{k}{a^{2}}=\frac{8\pi }{3}\rho \;,$$
(54)$$\frac{\ddot{a}}{a}=-\frac{4\pi }{3}\left(\rho +3p\right)\;,$$
where a dot represents the derivative with respect to the cosmic time *t*. The first Friedmann equation represents a constraint and the second equation determines the dynamics of the scale factor a(t).

However, Friedmann equations cannot be integrated in this form because they constitute a system of two differential equations for three unknowns, namely, the scale factor a(t), the density ρ(t), and the pressure p(t). So, it is necessary to add an equation to close the system. In the standard cosmological model, it is assumed that the perfect fluid is barotropic, i.e., it satisfies the equation of state
(55)p=wρ,
where *w* is the constant barotropic factor. In this case, Friedmann equations can be integrated in general for any value of *w*. To this end, instead of the second Friedmann equation, it is convenient to consider the conservation law for the energy-momentum
(56)$$T_{\;\;;\nu }^{\mu \nu }=0\;,$$
which, in the case of a barotropic perfect fluid, reduces to the equation
(57)$$\dot{\rho }+3\frac{\dot{a}}{a}\left(1+w\right)\rho =0\;.$$
This equation can be integrated and yields
(58)$$\rho =\rho_{0}a^{-3(1+w)}\;,$$
where $\rho_{0}$ is an integration constant. Furthermore, the remaining Friedmann equation (Equation 53) leads to
(59)$${\dot{a}}^{2}+k=\frac{8\pi }{3}\rho_{0}a^{-(1+3w)}\;.$$
In turn, the last equation can be integrated in a parametric form, leading to an explicit expression for the scale factor a(t), which depends also on the values of *k* and *w*.

In fact, only certain values are suitable for describing a given Einstein–de Sitter universe, simply characterized by the dominance of one given fluid. For instance, in a radiation-dominated universe, one can select w=1/3 or, analogously, w=0 for baryonic and dark-matter-dominated universes. Consequently, the phase dominated by the cosmological constant yields w≃−1. All these options can be exploited within the context of GTD, choosing the suitable set of free constants.

## 6. Geometrothermodynamic Cosmological Models

An important ingredient of the standard cosmological model described in the previous section is the equation of state (Equation 55) because it allows closing the system of differential equations. In GTD, we reach the same result in a different way.

The idea of geometrothermodynamic cosmology consists in applying Assumption 4 that the Universe is a thermodynamic system, implying that there should exist a fundamental equation from which we should derive all the thermodynamic properties of the system. As mentioned above, some of the fundamental equations derived in Section 4 have been used in the context of cosmology to construct specific models of dark matter and energy. In this section, we will consider the fundamental Equation (Equation 49), with Φ=S, $E^{1}=U$, and $E^{2}=V$, to integrate Friedmann equations. Then, in this case, the basic equations of geometrothermodynamic cosmology are Equations (Equation 53) and (Equation 54) plus the condition
(60)$$S=c1\ln\left(U+\frac{\alpha }{V}\right)+c_{2}\ln\left(V-\beta \right)\;,$$
where *S* is the entropy, *U* is the internal energy, and *V* is the volume. Then, we energy density should be given as $\rho =\frac{U}{V}$. Moreover, the fundamental Equation (Equation 60) should satisfy the first law of thermodynamics
(61)$$dS=\frac{1}{T}dU+\frac{p}{T}dV\;,$$
which also determines the equilibrium conditions
(62)$$\frac{1}{T}=\frac{\partial S}{\partial U}\;,\;\;\frac{p}{T}=\frac{\partial S}{\partial V}\;.$$
Notice that the equations of state, which relate intensive and extensive thermodynamic variables, can be derived from the equilibrium conditions (Equation 62). Thus, in geometrothermodynamic cosmology, the equations of state are a consequence of Assumption 4.

To investigate the cosmological models that can be derived from the Friedmann equations and (Equation 60), let us first consider the particular case with α=β=0. Then, the equations of state read
(63)$$\frac{1}{T}=\frac{c_{1}}{U}\;,\;\frac{p}{T}=\frac{c_{2}}{V}\;.$$
The first equation of state determines the thermodynamic temperature $T=\frac{U}{c_{1}}$, whereas the second equation of state can be written as $p=\frac{c_{2}}{c_{1}}\frac{U}{V}=\frac{c_{2}}{c_{1}}\rho$. This means that we are practically dealing with a barotropic equation of state with barotropic factor $w=\frac{c_{2}}{c_{1}}$. Consequently, the particular model with α=β=0 is equivalent to the standard cosmological model of relativistic cosmology.

### 6.1. More on Dark Energy

The above treatment does not fix univocally the GTD fluid. Specifically, it is always possible to slightly modify the fundamental equation in order to obtain more complicated or simply alternative cosmological models. As an example, we here derive a distinct GTD fluid responsible to speed up the Universe through another relevant choice of the fundamental equation. In particular, the key requirement is that our fluid exhibits a negative pressure that, as a working hypothesis, is proportional to the volume occupied by the fluid itself. This assumption provides a simple framework for identifying GTD fluids capable of accelerating the Universe today. However, it is important to note that there are certain considerations and limitations associated with this hypothesis, which we will clarify below. Hence, we have
(64)P=−kV,
where *k* is a constant, and *V* represents the volume of the universe. However, since the pressure of a fluid is given by
(65)$$P\equiv -\frac{\partial U(S,V)}{\partial V},$$
using Equation (Equation 64) into Equation (Equation 65), we determine the internal energy that reads
(66)$$U\left(S,V\right)=f\left(S\right)+k/2V^{2},$$

Notice that Equation (Equation 66) represents a fundamental equation that is consistent with our GTD approach mentioned earlier. However, despite its apparent simplicity, this scenario leads to evident thermodynamic instabilities. Indeed, this can be observed as the internal energy is a combination of two functions, namely the first, depending solely on entropy, and the second, depending on the Universe volume.

Consequently, the second-order crossed derivatives continually vanish, indicating the presence of thermodynamic instabilities. Similar conclusions can be sketched invoking a more general case, say P=−f(V), where f(V) is a generic function of volume. In other words, following the GTD recipe, a plausible and more robust choice might be under the form [61]
(67)P=−kVU(S,V),
in which, to characterize the large scale dynamics, one has to employ a pressure that is proportional to both the volume and the internal energy. Cumbersome algebra leads to
(68)$$U\left(S,V\right)=f\left(S\right)\;\exp\left(\frac{k}{2}\;V^{2}\right)\;.$$
Thus, considering the fundamental relation, Equation (Equation 68), and having Φ=U(S,V), $E^{1}=V$ and $E^{2}=S$, once computed the constant scalar curvature, we infer that f(S) might be a second-order polynomial in *S*, and so, invoking the Universe to be adiabatic, we write
(69)$$U=U_{0}\exp\left(c_{1}\;S+c_{2}\;V^{2}\right)\;,$$
where $c_{1}$ and $c_{2}$ are constants to determine.

Here, dark energy effects can be reobtained, and in fact, we have
(70)$$g_{U}^{♮}=\frac{c_{1}^{2}}{2\;c_{2}\;V^{2}}\;dS⊗dS+2\;\frac{c_{1}}{V}\;dS⊗dV+\frac{\;(1+2\;c_{2}\;V^{2})}{V^{2}}\;dV⊗dV\;.$$

From the perspective of GTD, we interpret this result by observing that Equation (Equation 69) corresponds to a system with a constant thermodynamic interaction. We raise the question of whether such systems can describe cosmological solutions.

To do so, it follows that
(71)$$T=c_{1}\;U_{0}\;e^{c_{1}\;S+c_{2}\;V^{2}}\;,\;P=-2\;c_{2}\;V\;U_{0}\;e^{c_{1}\;S+c_{2}\;V^{2}}\;,$$
or alternatively
(72)$$T=\;c_{1}\;U\;,\;P=-2\;c_{2}\;V\;U\;.$$
From Equation (Equation 72), we can observe that in order to have positive temperature and negative pressure, the constants $c_{1}$ and $c_{2}$ must be positive. Alternatively, for clarity, we can introduce the dark energy density, denoted as $\rho_{DE}=U/V$, and calculate the corresponding barotropic factor:(73)$$\omega_{DE}\equiv \frac{P}{\rho_{DE}}=-2c_{2}V^{2}.$$

It is evident from Equation (Equation 73) that an increase in volume, at constant energy *U*, leads to an increase in pressure. This can be interpreted as follows: at small volumes, the negative pressure decreases, indicating an expanding but non-accelerating universe. However, at larger volumes, the negative pressure becomes significant and contributes to the dynamics of the universe, eventually causing the observed late-time acceleration.

Thus, the effects of dark energy can be mimicked by our GTD fluid by satisfying the basic requirements of GTD, specifically with the constant thermodynamic interaction approach.

Assuming an adiabatic expansion of the universe, which is a common assumption in cosmology, and considering the universe as an isolated system, we can rewrite the fundamental Equation (Equation 69) in terms of the redshift (*z*) and calculate the evolution of the thermodynamic dark energy quantities with respect to *z*.

Assuming a constant entropy, the DE density corresponding to our GTD fluid can be expressed as:(74)$$\rho_{DE}=\exp\left(c_{1}S_{0}+\frac{c_{2}}{{\left(1+z\right)}^{6}}\right){\left(1+z\right)}^{3},$$
where we have assumed that the volume is given by $V∼a^{3}$, with $a={\left(1+z\right)}^{-1}$.

Next, we can calculate the barotropic factor (Equation 73) as a function of *z*, which can be rewritten as:(75)$$\omega_{DE}=-\frac{2c_{2}}{{\left(1+z\right)}^{6}}.$$

This expression suggests that we can recover the ΛCDM value, ω=−1, near z=0 by choosing a value of $c_{2}$ approximately equal to 0.5.

### 6.2. An Example of Inflationary Fluid

Consider now the general case of the fundamental Equation (Equation 60) with α and β different from zero. The equations of state can be expressed as (for simplicity we consider here the case k=0) [46]
(76)$$T=\frac{U}{c_{1}}+\frac{\alpha }{c_{1}V}\;,$$
and
(77)$$P=\frac{c_{2}UV^{2}+\alpha \left[\beta c_{1}+\left(c_{2}-c_{1}\right)V\right]}{c_{1}V^{2}\left(V-\beta \right)}\;.$$
We define the energy density as ρ=U/V and parametrize the volume as a function of the scale factor as $V=V_{0}\;a^{3}$ so that the pressure becomes a function of ρ and *a*. We use the standard convention of cosmology that the scale factor at current time $t_{0}$ is $a(t_{0})=1$, and thus $V_{0}$ can be understood as the volume of the Universe at current time.

Furthermore, we integrate the continuity equation and obtain
(78)$$\rho =K\frac{{\left(a^{3}V_{0}-\beta \right)}^{-\frac{c_{2}}{c_{1}}}}{a^{3}}-\frac{\alpha }{a^{6}V_{0}^{2}}\;,$$
where *K* is an integration constant, which together with α can be chosen such that the energy density is positive. Moreover, by fixing $c_{2}/c_{1}$, we can obtain in principle any polynomial dependence of the density of the kind
(79)$$\rho_{\inf}∼\frac{1}{a^{m}}\;.$$
By choosing $c_{2}/c_{1}$ appropriately, we can thus obtain a large number of models with inflationary behavior. It is also possible to achieve a period of strong expansion with an appropriate number of e-folds. For instance, consider the case $c_{2}/c_{1}=-8/9$, under the assumption that the constant β is small in the expression for the density (Equation 78). Then, expanding the first term for small values of β, we obtain
(80)$$\rho ≃\frac{KV_{0}^{8/9}}{a^{1/3}}-\frac{8\beta K}{9V_{0}^{1/9}a^{10/3}}-\frac{\alpha }{a^{6}V_{0}^{2}}\;.$$
If the first term is the dominating contribution to the density during the inflationary regime, it can produce the appropriate amount of e-foldings. Indeed, neglecting the last two terms in (Equation 80) for the duration of inflation, we can calculate the number of e-foldings
(81)$$\rho \left(a\right)≃\frac{KV_{0}^{8/9}}{a^{1/3}}\equiv \rho_{\inf}\left(a\right)$$
as follows. The integration of the first Friedmann equation yields the scale factor and the Hubble parameter
(82)$$a={\left(\frac{2\pi }{27}KV_{0}^{8/9}\right)}^{1/6}t^{6}\;,\;\;H=\frac{6}{t}\;.$$

The number of e-foldings can be calculated as
(83)$$N=\int Hdt=6\ln\left(\frac{t_{f}}{t_{i}}\right)\;,$$
where $t_{i}$ and $t_{f}$ are the times of the beginning and end of inflation, usually estimated to be $t_{i}=10^{-36}\;s$ and $t_{f}=10^{-32}\;s$; in this case, we obtain N=6(−32+36)ln10≃55, which is an appropriate number of e-foldings.

Since we are assuming that the density is dominated by the first term given in (Equation 80), we also obtain constraints on the possible values of the constants α and β. For instance, by requiring that the absolute value of each of the two additional terms in (Equation 80) is much smaller than the absolute value of the first term, i.e.,
(84)$$\begin{array}{ccc} & & \left|\alpha \right|\ll KV_{0}^{26/9}a_{i}^{17/3}=:\alpha_{c}\;, \end{array}$$
(85)$$\begin{array}{ccc} & & \left|\beta \right|\ll \frac{9}{8}\;V_{0}\;a_{i}^{3}=:\beta_{c}\;, \end{array}$$
where $a_{i}=a\left(t_{i}\right)$. From the fundamental Equation (Equation 60), it follows that the constant β should be related with some characteristic volume. Therefore, we assume that β is positive. On the other hand, α could, in principle, be both positive and negative. However, if we choose a negative α and assume that the two terms proportional to α and β cancel each other exactly at the beginning of inflation, we end up with the condition
(86)$$\left|\alpha \right|=\frac{8}{9}\beta KV_{0}^{17/9}a_{i}^{8/3}\ll \alpha_{c}\;.$$
This means that inflation starts off very cleanly, because the density at the beginning of inflation is determined by the inflationary term only. Then, the entire dynamics will be determined in terms of a small parameter
(87)$$\epsilon =\frac{\beta }{\beta_{c}}=-\frac{\left|\alpha \right|}{\alpha_{c}}\;.$$
Moreover, the present inflationary model contains two additional parameters, namely, $V_{0}$ and *K*.

In the present model, inflation lasts for about 55 e-foldings, i.e., during inflation the Universe expands for a factor of roughly $e^{55}≃7\cdot 10^{23}$ times. After that, during the periods of radiation and matter dominance, the Universe grows for further $10^{30}$ times. Furthermore, the current observable Universe has a diameter of about $l_{0}≃10^{26}\;m$, i.e., a volume of about $V_{0}≃10^{78}\;m^{3}$. Therefore, the diameter of the Universe at the beginning of inflation was $l_{i}=l_{0}/\left(10^{30}e^{55}\right)≃10^{-28}\;m$. Using $V=V_{0}a^{3}$ and the convention $a(t_{0})=1$, we can thus determine the scale factor at the beginning of inflation as $a_{i}=l_{i}/l_{0}≃10^{-54}$. Combining these numbers, we obtain
(88)$$\beta_{c}=\frac{9}{8}V_{0}a_{i}^{3}≃10^{-84}\;m^{3}\;,$$
i.e., $\beta_{c}$ equals about the volume of the Universe at the onset of inflation. This is a small number, but still much larger than the Planck volume, $l_{p}^{3}≃10^{-105}\;m^{3}$. We now fix the value of *K* for the estimated parameter $\alpha_{c}$. This can be achieved by requiring that the energy scale at the onset of inflation was of the order of the GUT scale of about $10^{16}\;\mathrm{GeV}$,
(89)$$\rho_{\inf}\left(t_{i}\right)=\frac{KV_{0}^{8/9}}{a_{i}^{1/3}}=\frac{10^{16}\;\mathrm{GeV}}{l_{i}^{3}}≃\frac{10^{6}J}{10^{-84}\;m^{3}}=10^{89}\;\frac{J}{m^{3}}\;.$$
The constant *K* then is
(90)$$K=\frac{\rho_{\inf}\left(a_{i}\right)a_{i}^{1/3}}{V_{0}^{8/9}}≃2\cdot 10^{2}\;{\mathrm{Jm}}^{-17/3}\;.$$
The peculiar unit of *K* is owed to the requirement that the inflationary density has the unit of energy density, and leads to
(91)$$\alpha_{c}≃10^{-78}\;{\mathrm{Jm}}^{3}\;.$$

We now analyze the value of $c_{2}/c_{1}$ for different amounts of e-foldings. In fact, we fixed the value of $c_{2}/c_{1}$ in order to obtain a particular number of e-foldings, but now we have to determine if this is the only possible choice. In order to investigate this question, we use the expression for the density (Equation 78), without specifying $c_{2}/c_{1}$, and expand it for small values of the parameter β. Neglecting the α– and β–terms, the inflationary density in the general case results in
(92)$$\rho_{\inf}≃\frac{k}{V_{0}^{c_{2}/c_{1}}}\frac{1}{a^{m}}\;,\;m=3+3\frac{c_{2}}{c_{1}}\;.$$
Using (Equation 83) for the number of e-foldings, we can calculate the values of *m* and *N* for a range of choices of $c_{2}/c_{1}$. Fixing $c_{2}/c_{1}{|}_{1}=-0.912$ results in a power-law dependence of the density of $m_{1}≃0.263$, slightly smaller than our previous choice, thus leading to stronger expansion with a number of e-foldings of $N_{1}≃70$. In contrast, choosing $c_{2}/c_{1}{|}_{2}=-0.898$ leads to $m_{2}≃0.307$ and the corresponding inflationary expansion of $N_{2}≃60$ e-foldings. Ultimately, defining $c_{2}/c_{1}{|}_{3}=-0.877$ yields the value of $m_{3}≃0.368$ and thus a number of e-foldings of $N_{3}≃50$.

Our previously chosen value of $c_{2}/c_{1}=-8/9$ corresponding to m=1/3 leads to N≃55 and lies somewhere between $N_{2}$ and $N_{3}$. We thus see that, although the equation of state parameter does not exactly have to be $c_{2}/c_{1}=-8/9$, there is still some room for variation. Small alterations of $O(10^{-2})$ in the value of $c_{2}/c_{1}$ lead to sizable fluctuations of O(10) in the number of e-foldings *N*, a crucial parameter in the description of inflation. The equation of state parameter is thus constrained in the regime of about $c_{2}/c_{1}\in \left[-0.912,-0.877\right]$ in order for the model to work.

As for the consequences on the dynamics, we do not expect qualitative changes when varying the equation of state parameter in this regime. Naturally, the values of the constants of the model, such as α, β, or *K*, will slightly change accordingly, as well as the evolution of the thermodynamic variables, since they depend on the choice of $c_{2}/c_{1}\in \left[-0.912,-0.877\right]$, but the qualitative features are preserved.

Using the expressions for the volume $V=V_{0}a^{3}$ and the density (Equation 78) for $c_{2}/c_{1}=-8/9$, and carrying out an expansion for small β, the temperature (Equation 76) becomes
(93)$$T\left(a\right)≃\frac{K}{c_{1}}a^{8/3}\left(V_{0}^{17/9}-\frac{8\beta V_{0}^{8/9}}{9}\frac{1}{a^{3}}\right)\;.$$
Furthermore, using the definition (Equation 87) for ϵ, and rewriting the scale factor as multiples of its value at the onset of inflation,
(94)$$a\left(t\right)=a_{i}x\;,$$
the temperature can be reexpressed as
(95)$$T\left(x\right)≃\frac{K}{c_{1}}V_{0}^{17/9}a_{i}^{8/3}x^{8/3}\left(1-\epsilon x^{-3}\right)\;.$$
We can assume that $c_{1}>0$ without loss of generality. Then, the temperature is positive as long as the expression in the bracket is positive, i.e., as long as *x* keeps growing and ϵ is small. However, this is one of the conditions which ensures the dominance of the inflationary term (85). Consequently, the same condition that must be satisfied in order to have clean inflation also guarantees that the temperature is positive at all times during inflation.

We now investigate the behavior of the pressure. From the general expression (Equation 77) and the density (Equation 78) for the special inflationary case $c_{2}/c_{1}=-8/9$ and small values of β, we obtain
(96)$$P\left(a\right)≃-\frac{\alpha }{a^{6}V_{0}^{2}}-\frac{8K}{9}V_{0}^{8/9}a^{-1/3}-\frac{8\beta K}{81}V_{0}^{-1/9}a^{-10/3}\;.$$
Using (Equation 87) and (Equation 94), we obtain
(97)$$P_{inf}\left(x\right)=\frac{\rho_{\inf}\left(a_{i}\right)}{x^{1/3}}\left[-\frac{8}{9}-\frac{\epsilon }{9}x^{-3}+\epsilon x^{-17/3}\right]\;.$$

It can be observed that for small values of ϵ, the pressure closely resembles that of a cosmological constant with a barotropic factor $\omega_{inf}=-8/9$. During inflation, the variable *x* increases, which implies that the terms proportional to ϵ dilute much faster compared to the dominant dependence on *x*. As a result, the pressure remains consistently negative throughout inflation, ensuring the appropriate expansion rate.

For the sake of comprehensiveness, it is important to note that the inflationary scenario described, while conceptually well constructed, does not dominate over radiation. In fact, it appears to be subdominant compared to radiation and, therefore, may not be suitable for describing inflation in a more realistic mixture of fluids, such as radiation, matter, and the GTD fluid.

This does not imply that the model itself cannot generate inflation, but rather that the specific fluid required for inflation would need to be fine-tuned, making it a less probable approach for inflation without scalar fields. However, further investigations are necessary, particularly regarding small perturbations. In the standard theory of inflation, it is predicted that scalar fields provide the small perturbations necessary for the formation of overdensities that eventually lead to the formation of observed structures.

In the context of GTD, in order to generate overdensities after an inflationary stage, a further phase transition that introduces inhomogeneities would be necessary. The details and feasibility of such a phase transition are currently under debate and remain unclear. This topic requires further exploration in future studies, yielding open the chance to have double transition-induced inflation and small perturbations.

## 7. Final Outlooks and Perspectives

In this work, we have provided a comprehensive overview of the formalism of GTD and its applications in the context of relativistic cosmology. We began by reviewing the fundamental concepts of classical thermodynamics, emphasizing the principle that the choice of thermodynamic potential does not affect the physical properties of thermodynamic systems. This crucial property serves as a guiding principle in formulating the geometric framework of GTD.

We first demonstrated that a change of potential in classical thermodynamics is achieved through the application of Legendre transformations to a specific seed potential. A significant step in this process is representing Legendre transformations as coordinate transformations in a (2 n + 1)-dimensional differential manifold, known as the phase space T, where *n* represents the number of thermodynamic degrees of freedom of the underlying system. This approach enables us to introduce additional geometric structures in T that remain invariant under Legendre coordinate transformations.

In particular, we introduced the concept of the contact 1-form Θ, which endows T with a Legendre invariant contact structure. This contact structure is found to be intimately connected to the first law of thermodynamics. Additionally, we introduced a Riemannian metric structure *G* in T and demanded it to be Legendre invariant as well.

As a result, we obtain a set of three different families of metrics, two of them being invariant under total transformations and the third one under partial transformations. In this way, the phase space is a Riemannian contact manifold that essentially contains in a geometric and invariant way the information about the fact that classical thermodynamics is invariant under Legendre transformations.

Furthermore, we introduce the concept of equilibrium space E as a subspace of T, which are related by means of a smooth map φ. It turns out that this map can also be used to induce the first law of thermodynamics in E, a fundamental equation for a thermodynamic system, and a set of three metrics that inherit the Legendre invariance property of the metrics *G* of T. The explicit form of the metrics *g* turns out to depend uniquely on the form of the fundamental equation. This means that the geometric properties of the equilibrium space depend on the particular fundamental equation induced by the embedding map φ.

We also propose a method to generate fundamental equations that consist in demanding that the map φ is harmonic. This implies that the subspace E is embedded in T as an extremal subspace, i.e., the volume of E is an extremal in T. As a result, we obtain a set of differential equations, equivalent to the Nambu–Goto equations of string theory, whose solutions can be interpreted as fundamental equations. In the particular case of systems with two thermodynamic degrees of freedom, we find some particular solutions that can be used to construct models in relativistic cosmology. Consequently, the essence of geometrothermodynamic cosmology consists in using fundamental equations derived from GTD as additional equation that allows us to integrate Friedmann equations of relativistic cosmology.

We have investigated the physical properties of particular fundamental equations, which represent the entropy of a thermodynamic system that explicitly depends on the internal energy and volume. In addition, they contain real parameters that enter the corresponding equations of state. Furthermore, if we assume that these equations of state can be applied to the entire Universe, we construct cosmological models that describe its evolution. In the first cosmological mod, the particular case in which α=β=0 turns out to be equivalent to the standard ΛCDM paradigm. On the other hand, the resulting cosmology with α≠0 and β≠0 has been shown to reproduce the main features of inflation, namely, the number of e-foldings (N≈55), which is consistent with commonly assumed parameters as the initial time ($t_{i}\approx 10^{-36}\;s$) and the final time ($t_{f}\approx 10^{-32}\;s$). Moreover, this inflationary model fixes the value $c_{2}/c_{1}=-8/9$ and demands that the parameters α and β be small. Evaluating these parameters shows that β corresponds to the volume of the Universe at the beginning of inflation and turns out to be ≈$10^{-90}\;m^{3}$. On the other hand, the ratio α/β determines the internal energy of the Universe at the beginning of inflation. The interaction constant α turns out to be small and ≈$10^{-78}\;{\mathrm{Jm}}^{3}$. These properties can be considered predictions of our cosmological model. This approach is general, but clearly represents a toy model toward the determination of more accurate inflationary fields. Indeed, the model itself appears subdominant with respect to radiation, being unable to drive the inflationary stage as radiation dominates.

In conclusion, we can say that GTD can be used in the framework of relativistic cosmology to construct valid cosmological scenarios. The particular case analyzed in this work describes an initial inflationary era and then reproduces the results of the standard cosmological model. Of course, it is necessary to further investigate the details of the models generated in the framework of geometrothermodynamic cosmology to determine if they are able to describe other important features such as smooth transitions between the specific eras and cosmological perturbations. These are tasks for future works.

## Figures and Tables

**Figure 1 entropy-25-01037-f001:**
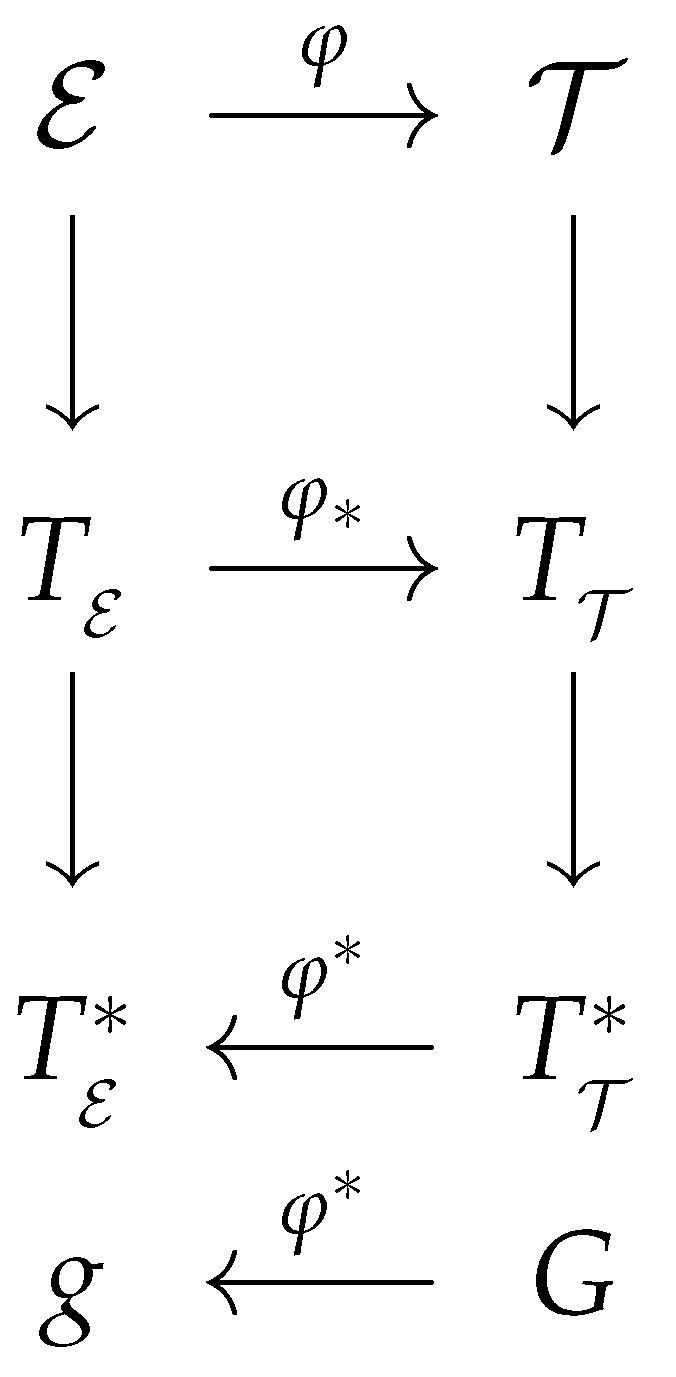
Geometric structure of GTD.

## Data Availability

Not applicable.

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
