# Peer review of "Geometrothermodynamic Cosmology"

_entropy, 2023, doi:10.3390/e25071037_

Round 1

Reviewer 1 Report

The authors present a comprehensive review of geometrothermodynamics (GTD), a formalism connecting aspects of differential geometry with thermodynamic systems. The paper demonstrates a strong level of coherence, clarity, and consistency, building upon the authors' previous research.

In terms of cosmological implications, the paper explores the assumption that the Universe can be regarded as a thermodynamic system governed by a fundamental equation derived from a 2-dimensional GTD. This approach yields equations of state that provide descriptions for specific models of dark matter and dark energy. Additionally, the authors provide a concise comparison of the presented theoretical framework with observational data.

Given the strengths of this manuscript, I honestly recommend its publication without any further comments or suggestions.

Author Response

Dear editor,

we are grateful to the anonymous referee for her/his effort in evaluating our work and in particular for the encouraging words she/he spent for us. 

Once again we intend to acknowledge her/him. Our warmest regards, the authors

Reviewer 2 Report

In this paper the authors review the “geometrothermodynamic” approach to thermodynamics. The foundational property of this approach is to demand invariance under Legendre transformations, reflected by both the contact 1-form $\Theta$ and the thermodynamic geometry $G$. It is shown that together with the assumption of quasi-cohomogeneity of the fundamental equation, this severely restricts the form of possible thermodynamic metrics. There is also a proposal for deriving the fundamental equation from the Nambu-Goto action for the volume of the equilibrium state. This is shown to yield the “standard” equation of state of cosmology, including the inflationary epoch.

I find the present paper interesting. However, I have several important questions about the geometrothermodynamic approach.

11) The basic feature of this approach is to demand “Legendre transform invariance”, or as stated on page 7 below Eq. (2) “the properties of the system do not depend on the choice of the potential $\tilde \Phi$ [5]”.

I find this statement/requirement very puzzling, as it is well known that thermodynamic systems feature very different thermodynamic behavior in different thermodynamic ensembles (related by Legendre transform). Considering black holes in AdS as an example: it is well known that in the grand-canonical ensemble these admit Hawking-Page phase transition, whereas in the canonical one they demonstrate Van der Waals-like phase transitions – how is this behavior independent of the ensemble studied?

In any case, some explanations are due in here.

22) What is the physical meaning of the thermodynamic geometry $G_{AB}$?

33) I am not sure what the arrows in Eqs. (6) and (7) are supposed to mean. Are $\tilde \Theta$ and $\tilde G$ the same invariant geometric object as $\Theta$ and $G$? If so, do we just require the “form invariance”, as is the standard case with canonical transformations?

44) Is $k$ in (10) an arbitrary real parameter?

55) I did not understand why in Sec. III suddenly the curvature scalar for the derived thermodynamic geometries $g_{ab}$ is studied – is this supposed to capture some critical behavior of the system? If so, why do we only focus on Ricci scalar, and not other invariants? What about “non-curvature” singularities?

66) In Sec. IV the dynamics in the phase space is proposed, which supposedly leads to fundamental equation. Is there any reason as to why this dynamics is identified with the Nambu-Goto equations of motion (governing the free motion of higher-dimensional objects)? Could we impose some other dynamics?

77) Is there any physical intuition as to why fundamental equation (49) derived in the previous manner should describe equation of state for our Universe? (Are we for example saying that our universe is a brane “floating” in the higher-dimensional thermodynamic phase space?

I I believe these questions need to be clarified before publishing this paper.

Author Response

Dear Editor, 

we hereafter attach our answer to Referee no. 2 and we are grateful for her/his great help in improving our manuscript. Regards, the authors.

Reviewer 3 Report

The paper, ``Geometrothermodynamic cosmology,'' reviews a Riemannian geometry formulation of thermodynamics known as geometrothermodynamics. Upon providing an action for the metric on the Riemannian `thermodynamics manifold', it is shown what field equations the thermodynamics variables follow. Solutions are derived for simple systems, which have applications to cosmology. Indeed, it is shown how these solutions can reproduce standard barotropic equations of state, but also new dark energy or inflationary-like fluids.   The paper is clear, well written, and interesting. It reviews well the topic for the non-expert readers, and it presents the latest results in the field. I thus might recommend publication if the authors can addressed the following:   \begin{enumerate} \item In the last paragraph before Section 6, it is said that ``only certain values are in agreement with observations'' when referring to the equations of state of radiation, matter, and dark energy. I would be more careful with the phrasing there. Observations play in role in determining what amount of radiation, matter, and dark energy there is, but they do not determine the individual values of $w$. Observations can tell us something about the `effective' or `total' equation of state of the universe, by inference through the observed expansion rate of the universe, but the individual equations of state are really derived from microphysical considerations. Since a dust fluid is like particles at rest, one can derive $w=0$ (no pressure); since radiation is like a photon bath, one can derive $w=1/3$; etc. \item Somewhat related to the above, how would the authors respond to the fact that their method only provides thermodynamics relations, which do not tell us anything about the fundamental (microphysical) origin of these thermodynamics relations? For instance, they can reproduce a barotropic equation of state with $w=c_2/c_1$, but the parametrization leaves $c_1$ and $c_2$ free. Therefore, one can postulate any $w$, regardless of the physical origin of such matter. Similarly, if a `new' equation of state is found for dark energy, this still does not tell us what dark energy is. \item Regarding the inflationary example, it is said that it does not work because the fluid remains sub-dominant to radiation. How can this be the case considering an inflationary `fluid' has a constant energy density that will eventually dominate over that of radiation (which redshifts as $\rho\propto a^{-4}$)? Is it because of the non-standard (non-linear equation of state) nature of the inflationary equation of state? If so, in what sense is it really inflationary? Can it solve the usual Big Bang problems inflation was invented for? And back to the issues raised above, if there is no microphysical description for that inflationary fluid, how can one explain and compute the origin of the primordial fluctuations, which lead to the cosmic microwave background? \end{enumerate}

Author Response

Dear Editor, 

we hereafter attach our answer to Referee no. 3 and we are grateful for her/his great help in improving our manuscript. Regards, the authors.

Round 2

Reviewer 2 Report

I thank the authors for their explanations and changes to the manuscript. As I already indicated in my previous report, I find the paper rather intriguing. While I am still puzzled by a few aspects of the presented theory (for example, the “restriction” to one statistical ensemble is really puzzling to me, or the physical interpretation of the Nambu-Goto action in this context) I believe this should not prevent the publication of this paper. For these reasons I am happy to recommend the publication in the present form.

I thank the authors for their explanations and changes to the manuscript. As I already indicated in my previous report, I find the paper rather intriguing. While I am still puzzled by a few aspects of the presented theory (for example, the “restriction” to one statistical ensemble is really puzzling to me, or the physical interpretation of the Nambu-Goto action in this context) I believe this should not prevent the publication of this paper. For these reasons I am happy to recommend the publication in the present form.

Reviewer 3 Report

The authors have appropriately answered my questions and addressed my concerns. The manuscript is suitable for publication.